**Subject Area:**
biochemistry/cellular biology/molecular biology

breast cancer, RNMT, PIK3CA, mRNA cap, PI3Kα, cell proliferation

**Author for correspondence:**
Victoria H. Cowling
e-mail: v.h.cowling@dundee.ac.uk

†Present address: AstraZeneca, CRUK Cambridge Institute, Robinson Way, Cambridge CB2 0RE, UK.
‡Present address: NDM Research Building, University of Oxford, Old Road Campus, Headington, Oxford OX3 7FZ, UK.

# Oncogenic PIK3CA mutations increase dependency on the mRNA cap methyltransferase, RNMT, in breast cancer cells

Sianadh Dunn†, Olivia Lombardi‡, Radoslaw Lukoszek and Victoria H. Cowling

Centre for Gene Regulation and Expression, School of Life Sciences, University of Dundee, Dundee DD1 5EH, UK

VHC, 0000-0001-7638-4870

Basic mechanisms in gene expression are currently being investigated as targets in cancer therapeutics. One such fundamental process is the addition of the cap to pre-mRNA, which recruits mediators of mRNA processing and translation initiation. Maturation of the cap involves mRNA cap guanosine N-7 methylation, catalysed by RNMT (RNA guanine-7 methyltransferase). In a panel of breast cancer cell lines, we investigated whether all are equivalently dependent on RNMT for proliferation. When cellular RNMT activity was experimentally reduced by 50%, the proliferation rate of non-transformed mammary epithelial cells was unchanged, whereas a subset of breast cancer cell lines exhibited reduced proliferation and increased apoptosis. Most of the cell lines which exhibited enhanced dependency on RNMT harboured oncogenic mutations in PIK3CA, which encodes the p110α subunit of PI3Kα. Conversely, all cell lines insensitive to RNMT depletion expressed wild-type PIK3CA. Expression of oncogenic PIK3CA mutants, which increase PI3K p110α activity, was sufficient to increase dependency on RNMT. Conversely, inhibition of PI3Kα reversed dependency on RNMT, suggesting that PI3Kα signalling is required. Collectively, these findings provide evidence to support RNMT as a therapeutic target in breast cancer and suggest that therapies targeting RNMT would be most valuable in a PIK3CA mutant background.

## 1. Background

Basic mechanisms in gene expression are currently being investigated as targets in cancer therapeutics [1,2]. In eukaryotes, gene expression is dependent on the mRNA cap, a structure that protects nascent transcripts from nucleases and recruits complexes which mediate mRNA processing, including splicing, export and translation initiation [3–5]. The mRNA cap forms during transcription and remains on the transcript throughout its lifetime. The basic cap structure is m7G(5′)ppp(5′)Xm (X denotes the first transcribed nucleotide), 7-methylguanosine linked to the first transcribed nucleotide via 5′ to 5′ triphosphate bridge. A series of enzymes catalyse cap formation initiating with RNGTT (RNA guanylyltransferase and 5′ phosphatase), which catalyses the addition of the inverted guanosine cap to the nascent transcript. Subsequently, RNMT (RNA guanine-7 methyltransferase) and its activating subunit, RAM (RNMT-activating miniprotein), are recruited to transcribing RNA pol II [6–8]. RNMT–RAM methylates the guanosine cap on the N-7 position to create the cap structure, m7G(5′)ppp(5′)X [9,10]. RNMT–RAM also has a major role in enhancing the transcription of the most active genes via interactions with protein complexes of the transcription machinery [11]. RNMT–RAM can also be involved in the process of recapping, during which partially capped or uncapped transcripts receive a cap in the cytoplasm [12].

royalsocietypublishing.org/journal/rsob    Open Biol. **9**: 190052

Thus, RNMT–RAM has a potent role in gene expression, promoting transcription and translation.

Cellular signalling pathways can regulate mRNA cap formation on specific target mRNAs, resulting in the regulation of their expression. For example, c-Myc and E2F1 (and potentially other transcription factors) increase RNA pol II phosphorylation which promotes mRNA cap formation via recruitment of the capping enzymes [13–17]. c-Myc-dependent mRNA cap methylation also requires upregulation of SAHH, the enzyme which hydrolyses the inhibitory bi-product of methylation reactions [18]. Upregulation of mRNA cap formation is critical for c-Myc function; inhibition of this process is synthetic lethal with deregulated c-Myc [15,18]. Formation of the mRNA cap is also regulated during the cell cycle by CDK1-dependent phosphorylation and activation of RNMT [19]. As CDK1 and c-Myc are implicated in tumorigenic events and cap-dependent mRNA translation is deregulated in many cancers, this raises the potential of utilizing the mRNA capping enzymes as therapeutic targets to selectively inhibit protein synthesis in cancer cells [20,21]. In particular, because RNMT ligands have low polarity, there may be an opportunity to create active site inhibitors with a low charge that are able to cross the plasma membrane [22].

When considering RNMT as a therapeutic target, is it valuable to determine which oncogenes drive a heightened state of dependency on RNMT. This gives an important indication of which cancer types and patient populations are most likely to response to RNMT inhibitors. In this study, we investigated the impact of RNMT depletion on the proliferation of a panel of breast cancer cell lines. We found that cells with activating mutations in PIK3CA, which are present in around 35% of breast cancers, exhibit enhanced dependency on RNMT for proliferation [23]. Cells expressing wild-type (WT) PIK3CA had their dependency on RNMT increased by the exogenous expression of oncogenic PIK3CA mutants, which increase PI3Kα activity. Conversely, cells expressing oncogenic PIK3CA mutations had their enhanced RNMT dependency reversed by the use of low doses of PI3Kα inhibitors, suggesting that PI3Kα signalling is mediating this dependency. Taken together, these data demonstrate that breast cancer cells with activating mutations in PIK3CA are selectively sensitive to RNMT inhibition.

# 2. Material and methods

## 2.1. Tissue culture

Cells were cultured at 37°C and 5% $CO_2$. The carcinoma cells were purchased from the American Type Culture Collection (ATCC) and IMECs (immortalized normal mammary epithelial cell lines) kindly provided by Dr James DiRenzo (Dartmouth Medical School). Experiments were typically performed within four weeks of culture. Breast carcinoma cell lines were maintained in RPMI, 10% FBS, 2 mM L-glutamine and 1% Anti-anti. IMEC cells were maintained in DMEM/F12, 2 mM L-glutamine, 5 mg ml$^{-1}$ insulin, 10 ng ml$^{-1}$ EGF, 0.5 µg ml$^{-1}$ hydrocortisone and 1% Anti-anti. pBMN-IRES neo-constructs containing Myc-PIK3CA, RNMT-GFP and mutants or empty vector control were transduced into cells by retro-viral infection and selected using 0.5 mg ml$^{-1}$ G418 for 7 days.

## 2.2. PIK3CA sequencing

RNA was extracted using the GeneJET RNA Purification Kit (ThermoFisher Scientific). Approximately 5–10 µg of RNA was digested with 2 µl of Turbo DNAse (ThermoFisher Scientific) to remove genomic DNA and purified using the RNA Clean & Concentrator Kit (Zymo Research), and 1 µg of RNA was used in the reverse transcription reaction using RevertAid H Minus Reverse Transcriptase (ThermoFisher Scientific) and the following RT primer: tcaaaccctgtttgcgttta. About 2 µl of reverse transcription reaction was used as a template in the PCR with KOD Hot Start DNA Polymerase (Merck) with primers PIK3CA_fwd2 tgctttgggacaaccataca and PIK3CA_rev3 cctatgcaatcggtctttgc. The PCR product was purified with the QIAquick PCR Purification Kit (Qiagen) and sequenced using the following primers: PIK3CA_FWD2 tgctttgggacaaccataca; 669R cacagtcatggttgattttcaga; 1166R cctgggattggaacaaggta; 1288F ggcatggggaaatataaacttg; 1664R tctcctgctcagtgatttcaga; 2143R gcctcgacttgcctattcag; 2312F caggcttgaagagtgtcgaa; 2636R gaactgcagtgcacctttca; PIK3CA_REV3 cctatgcaatcggtctttgc.

## 2.3. Cell transfection, treatment and counting

Cells were seeded at equivalent density, at approximately $2.5 \times 10^5$ cells per six-well plate, and immediately transfected with siRNAs (Dharmacon) using Lipofectamine RNAiMax transfection reagent. For a six-well dish, 0.16 nmol siRNA was transfected with 3 µl of Lipofectamine RNAiMax and 200 µl of serum-free DMEM. siRNAs used were siRNMT 1 (D-019525-01-0050), siRNMT 2 (019525-02-0050), siRNMT 3 (019525-03-0050) and non-targeting control (D001210-03-0050). When relevant, cells were treated with 0.1 µM staurosporine for 3 h prior to lysis and 50 nM GDC-0941 or 15 nM BYL719 for 72 h prior to lysis or cell counting. For experiments requiring longer than 48 h siRNA transfection, the growth media were replenished at 48 h. Cells were counted using a Countess cell counter (Life Technologies).

## 2.4. Generation of GFP-RNMT and PIK3CA mutant stable cell lines

HCC-1806 cells were retrovirally transduced to stably express siRNA-resistant, GFP-tagged full-length RNMT. ZR-751, IMEC and MDA-MB-231 cells were retrovirally transduced to stably express Myc-tagged oncogenic mutants of p110α (E545K, C420R and H1047R).

## 2.5. Statistical analysis

Statistical significance was assessed by one-way analysis (ANOVA) followed by Dunnett's multiple comparison tests using GraphPad PRISM 5.0. A value of $p \leq 0.05$ is denoted with *, $p \leq 0.01$ denoted with **, $p \leq 0.001$ denoted with ***.

## 2.6. Cell extract preparation

Cell lysis was performed at 4°C. Culture media were removed, cells were washed twice with ice-cold PBS and lysed in ice-cold F buffer, comprising 10 mM Tris (pH 7.05), 50 mM NaCl, 30 mM Na-pyrophosphate, 50 mM NaF, 5 µM $ZnCl_2$, 10% glycerol, 0.5% Triton X-100, 1 mM EGTA,

1 mM EDTA, and 1 mM sodium orthovanadate) supplemented with 0.1 TIU (trypsin inhibitor unit) aprotinin, 1 μM pepstatin, 10 μM leupeptin and 1 mM DTT immediately before use. For analysis of phosphorylated protein, lysis buffer was supplemented with Sigma Phosphatase Inhibitors (cocktail mixtures 2 + 3). Cell lysates were collected by scraping and the soluble fraction was collected following centrifugation at 16 000 $g$ for 10 min at 4°C. Protein concentration was determined using the Bradford method and extracts were normalized for protein content. Typically, 5–20 μg of cell extract was analysed. Band intensity was quantitated using Image J software.

## 2.7. Antibodies

Anti-RNMT, RAM and AKT antibodies were developed in-house and raised against full-length recombinant human proteins in sheep and sera purified against the antigen. Other antibodies purchased were Actin (Abcam-8226), PARP (CST 9541), AKT T308P (CST 9275), AKT S473P (CST 9271), 4E-BP1 Thr 37/46 (CST 9459), P-4EBP Thr 70 (CST 9455), 4E-BP (CST 9452), p70 S6 kinase Thr 389 (CST 9205), c-Myc (CST 9402) and p70 S6 kinase (CST 9202).

## 2.8. *In vitro* cap methyltransferase assay

0.25, 0.5 or 1 μg of cell extracts were incubated with 2 mM SAM, 20 U RNasin, MT buffer (10 mM Tris pH 8, 0.6 mM KCl, 0.125 mM MgCl$_2$) and *in vitro* transcribed $^{32}$P G-capped RNA at 37°C for 10 min. RNA was purified and resuspended in 4 μl of 50 mM Na-acetate (pH 5.5). RNA was P1 nuclease-treated to release free guanosine cap. GpppG (basic guanosine cap) and m$^7$GpppG (N7-methylated guanosine cap) resolved on PEI cellulose plates in 0.4 M ammonium sulfate, visualized by phosphoimager and quantified using AIDA IMAGER software.

# 3. Results

## 3.1. Breast cancer cell lines harbouring oncogenic PIK3CA exhibit enhanced dependency on RNMT

We investigated the proliferative response of a panel of breast cancer cell lines and a normal mammary epithelial cell line to a reduction in RNMT expression. Initially, a panel of eight breast cancer cell lines with a spectrum of mutations was analysed: MCF7, HCC1806, JIMT-1, T47D, BT-549, MDA-MB-231, CAMA-1 and ZR-75-1 (table 1). Cell lines were purchased from ATCC (American Type Culture Collection) and used within four to six weeks of culture to reduce passage-dependent effects. Known mutations of cancer-associated genes in these cell lines were extracted from the COSMIC database (table 1). In addition, a low-passage, non-transformed TERT-IMEC (TERT-immortalized mammary epithelial cell line) was analysed [24]. RNMT expression was reduced by transfection of three independent RNMT siRNAs and a non-targeting siRNA control. All cell lines harbouring PIK3CA-activating mutations (MCF7, JIMT-1 and T47D, marked with a red asterisk), and one cell line expressing WT PIK3CA (HCC-1806), exhibited reduced proliferation in response to transfection of all three RNMT siRNAs (figure 1a). By contrast, the proliferation of IMECs and other breast cancer cell lines expressing WT

PIK3CA (BT-549, MDA-MB-231, CAMA-1 and ZR-75-1) was unaffected by RNMT siRNA transfection. The PIK3CA coding region sequence was verified for all cell lines (electronic supplementary material, figure S1). Combining three independent experiments, the average cell number 96 h after RNMT siRNA 1 transfection was calculated relative to the cell number in a control transfection (figure 1b). In agreement with the growth curves in figure 1a, in response to RNMT siRNA transfection, there was a significant reduction in the cell number for MCF7, HCC-1806, JIMT-1 and T47D cell lines compared with IMEC. However, BT-549, CAMA-1, ZR-75-1 and MDA-MB-231 cell lines did not exhibit a significant loss in cell number following RNMT siRNA transfection. This difference in sensitivity to RNMT siRNA transfection did not correlate with the level of RNMT reduction. All cell lines exhibited an equivalent reduction in RNMT expression in response to transfection of all three RNMT siRNAs (figure 1c). As observed previously, expression of the RNMT-activating subunit, RAM, was also reduced in response to reduced RNMT expression (figure 1c) [10,25]. RNMT and RAM are co-translated, and their interaction protects the two proteins from proteasome-mediated degradation.

Since we observed a similar proliferative defect following transfection of three independent RNMT siRNAs, it is likely that this impact is specifically due to a reduction in RNMT levels. To strengthen this conclusion, an RNMT-GFP construct resistant to RNMT siRNA 1 was introduced into HCC1806 cells by retroviral infection. As expected, transfection of RNMT siRNA 1 inhibited endogenous RNMT expression but did not inhibit expression of RNMT-GFP (figure 2a). Consistent with figure 1, depletion of endogenous RNMT significantly inhibited the proliferation of control HCC1806 cells, but this effect was reversed by the expression of RNMT-GFP (figure 2b). This confirms that the proliferative defect observed following RNMT siRNA transfection is probably due to a reduction in RNMT expression and not a consequence of siRNA off-target effects

The reduction in cell number that we observed in response to reduced RNMT expression could be due, in part, to an increase in apoptosis. Inhibition of RNMT and RAM has been observed to induce apoptosis in HeLa cells, a cervical cancer cell line [10,26]. We analysed apoptosis in a subset of our breast cancer cell panel by detecting fragments of PARP, a nuclear antigen cleaved during apoptosis (figure 2c). All four cell lines investigated were competent to undergo apoptosis; all exhibited cleaved PARP in response to incubation with Staurosporine, a pro-apoptotic agent. Cell lines exhibiting enhanced dependency on RNMT (MCF7 and HCC-1806), also exhibited PARP cleavage in response to reduced RNMT expression, indicating that apoptosis was taking place. MDA-MB-231 and ZR-75-1 cells, which do not exhibit a proliferative defect in response to RNMT siRNA transfection, also did not exhibit PARP cleavage in response to reduced RNMT expression (figure 2c). These data demonstrate that RNMT depletion induces a cytotoxic response in RNMT-dependent breast cancer cell lines.

## 3.2. RNMT dependency does not correlate with RNMT expression, activity or rate of protein synthesis

To investigate the molecular mechanisms governing dependency on RNMT, we investigated whether cells sensitive to

**Table 1.** Cancer-associated mutations in the breast cancer cell lines. Information was obtained from the online Catalogue of Somatic Mutations In Cancer (COSMIC) database. Only genes defined as being cancer associated by COSMIC are listed.

| cell line | gene | transcript | AA mutation | CDS mutation | type |
|---|---|---|---|---|---|
| MCF7 breast, carcinoma, ER-PR-positive carcinoma | ATP2B3 | ENST00000359149 | p.V882E | c.2645T>A | substitution—missense |
| | EP300 | ENST00000263253 | p.R1356* | c.4066C>T | substitution—nonsense |
| | ERBB4 | ENST00000342788 | p.Y1242C | c.3725A>G | substitution—missense |
| | FUS | ENST00000254108 | p.D502delD | c.1503_1505delGGA | deletion—in frame |
| | GATA3 | ENST00000379328 | p.D336fs*17 | c.1006_1007insG | insertion—frameshift |
| | KMT2C | ENST00000262189 | p.I3590 L | c.10768A>C | substitution—missense |
| | MAP3K13 | ENST00000424227 | p.D380N | c.1138G>A | substitution—missense |
| | MYH9 | ENST00000216181 | p.K682_L687delKLDPHL | c.2043_2060del18 | deletion—in frame |
| | PIK3CA | NM_006218.1 | p.E545 K | c.1633G>A | substitution—missense |
| HCC1806 breast, carcinoma, basal (triple-negative) carcinoma | TP53 | ENST00000269305 | p.T256fs*90 | c.? | frameshift |
| | PIK3CA WT | | | | |
| JIMT-1 breast (carcinoma; HER-positive carcinoma) | PIK3CA | NM_006218.1 | p.C420R | c.? | substitution—missense |
| | TP53 | ENST00000269305 | p.R248 W | c.? | substitution—missense |
| T47D breast, carcinoma, ductal carcinoma | ACVR1 | ENST00000263640 | p.N100D | c.298A>G | substitution—missense |
| | ARID1A | ENST00000324856 | p.Q944* | c.2830C>T | substitution—nonsense |
| | KDM5C | ENST00000375401 | p.Q920* | c.2758C>T | substitution—nonsense |
| | KDM5C | ENST00000375401 | p.L921 V | c.2761C>G | substitution—missense |
| | KMT2C | ENST00000262189 | p.I3590 L | c.10768A>C | substitution—missense |
| | MLLT4 | ENST00000366809 | p.T838fs*4 | c.2512_2513insA | insertion—frameshift |
| | PIK3CA | NM_006218.1 | p.H1047R | c.3140A>G | substitution—missense |
| | SPEN | ENST00000375759 | p.D329fs*2 | c.984_985insA | insertion—frameshift |
| | TP53 | ENST00000269305 | p.L194F | c.580C>T | substitution—missense |
| CAMA-1 breast, carcinoma, luminal carcinoma | CDH1 | ENST00000261769 | p.? | c.1712-1G>A | unknown |
| | PTEN | ENST00000371953 | p.D92H | c.274G>C | substitution—missense |
| | PTEN | ENST00000371953 | p.D268_F279>12 | c.802_837GACAAAATGTTCAC TTTTGGTAAATACATTCTTC>36 | complex—compound substitution |
| | PIK3CA WT | | | | |
| BT-549 breast, carcinoma, ductal carcinoma | PTEN | ENST00000371953 | p.V275fs*1 | c.823delG | deletion—frameshift |
| | TP53 | ENST00000269305 | p.R249S | c.747G>C | substitution—missense |
| | PIK3CA WT | | | | |

(Continued.)

royalsocietypublishing.org/journal/rsob   Open Biol. **9**: 190052

**Table 1.** (Continued.)

| cell line | gene | transcript | AA mutation | CDS mutation | type |
|---|---|---|---|---|---|
| MDA-MB-231 breast, carcinoma, basal (triple-negative) carcinoma | BRAF | ENST00000288602 | p.G464 V | c.1391G>T | substitution—missense |
| | CD79A | ENST00000221972 | p.C106Y | c.317G>A | substitution—missense |
| | CRTC3 | ENST00000268184 | p.P578A | c.1732C>G | substitution—missense |
| | NF2 | ENST00000338641 | p.E231* | c.691G>T | substitution—nonsense |
| | PDGFRA | ENST00000257290 | p.Y172F | c.515A>T | substitution—missense |
| | TP53 | ENST00000269305 | p.R280 K | c.839G>A | substitution—missense |
| | PIK3CA WT | | | | |
| ZR-75-1 breast, carcinoma | PTEN | ENST00000371953 | p.L108R | c.323T>G | substitution—missense |
| | PIK3CA WT | | | | |
| HCC38 breast, carcinoma, ductal | IKBKB | ENST00000520810 | p.A360S | c.1078G>T | substitution—missense |
| | TOP1 | ENST00000361337 | p.K326R | c.977A>G | substitution—missense |
| | TP53 | ENST00000269305 | p.R273 L | c.818G>T | substitution—missense |
| | PIK3CA WT | | | | |
| HCC1569 breast, carcinoma, HER-positive | TP53 | ENST00000269305 | p.E294* | c.? | substitution—nonsense |
| | TP53 | ENST00000269305 | p.S227fs*? | c.? | frameshift |
| | PIK3CA WT | | | | |
| NCI-H460 lung, carcinoma, large cell carcinoma | ARID1A | ENST00000324856 | p.I2135_L2136del | c.6403_6408delATTCTG | deletion—in frame |
| | KDR | ENST00000263923 | p.G1304C | c.3910G>T | substitution—missense |
| | KEAP1 | ENST00000393623 | p.D236H | c.706G>C | substitution—missense |
| | KRAS | ENST00000311936 | p.Q61H | c.183A>T | substitution—missense |
| | LIFR | ENST00000263409 | p.L493F | c.1477C>T | substitution—missense |
| | MYH11 | ENST00000338282 | p.D757E | c.2271C>A | substitution—missense |
| | NBN | ENST00000265433 | p.G224A | c.671G>C | substitution—missense |
| | PIK3CA | NM_006218.1 | p.E545 K | c.1633G>A | substitution—missense |
| | PPP2R1A | ENST00000322088 | p.E64D | c.192G>T | substitution—missense |
| | RAD21 | ENST00000297338 | p.D118 V | c.353A>T | substitution—missense |
| | TNFAIP3 | ENST00000237289 | p.R136C | c.406C>T | substitution—missense |
| | ZFHX3 | ENST00000268489 | p.P3675Q | c.11024C>A | substitution—missense |
| HCC1500 breast, carcinoma, ER-PR-positive | PIK3CA | NM_006218.1 | p.T1025T | c.3075C>T | substitution—coding |

royalsocietypublishing.org/journal/rsob Open Biol. 9: 190052

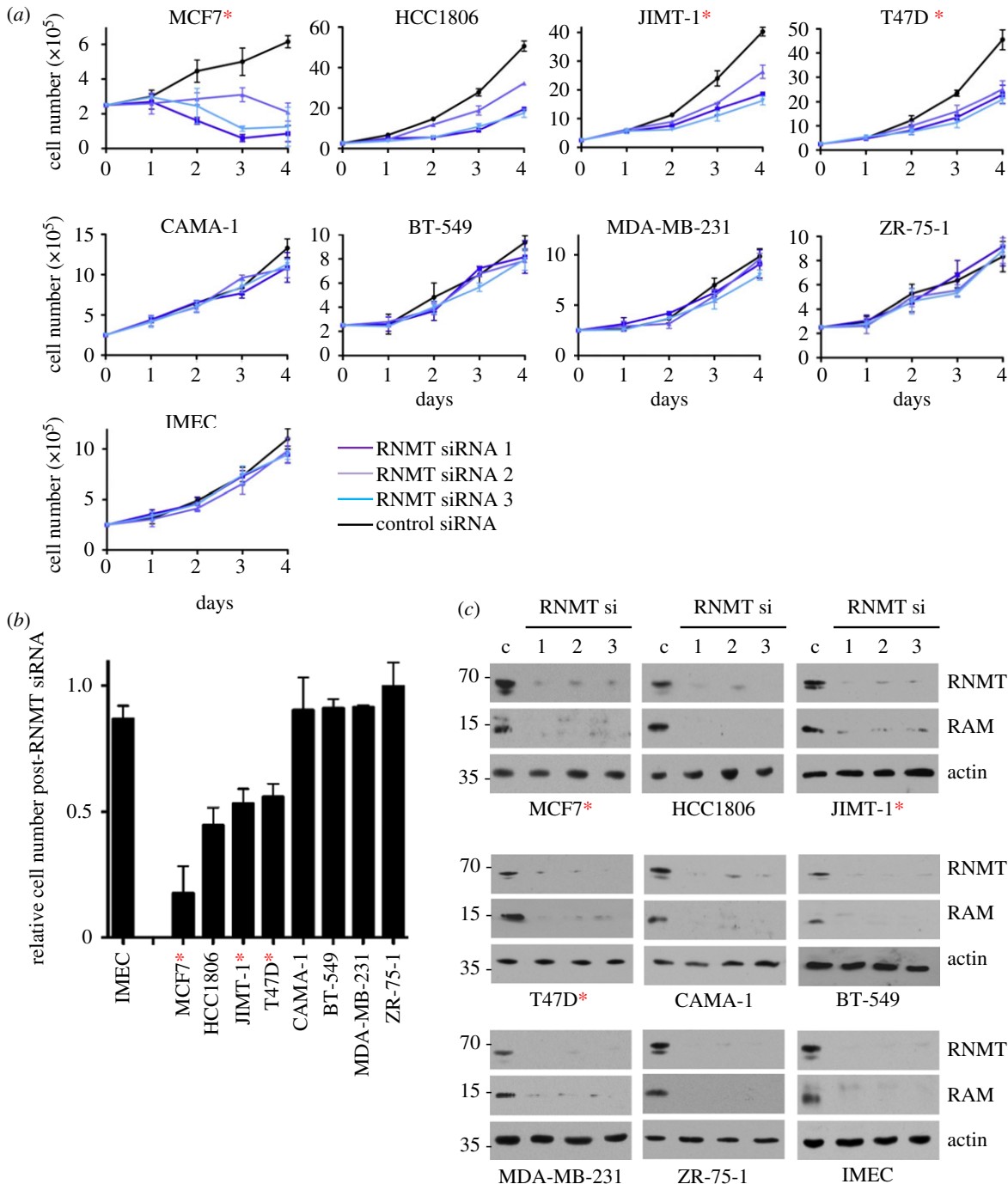

**Figure 1.** Breast cancer cell lines expressing PIK3CA mutants are sensitive to RNMT depletion. Breast cancer cell lines MCF7, HCC1806, JIMT-1, T47D, BT-549, MDA-MB-231, CAMA-1 and ZR-75-1, and a normal mammary epithelial cell line, IMEC, were transfected with three independent RNMT siRNAs or a non-targeting control siRNA. (a) Over 4 days, average cell number and standard deviation from three independent transfections are reported. (b) Ninety-six hours following transfection with RNMT siRNA1, the cell number was determined relative to cells transfected with non-targeting siRNA. Average result and standard deviation for three independent experiments are reported. Statistical significance was assessed by ANOVA followed by Dunnett's multiple comparison tests. '***' indicates a p-value of less than 0.001 compared with IMEC control. (c) Seventy-two hours post-transfection, expression of RNMT, RAM and actin was detected by western blot. Data presented are one out of two independent experiments with consistent results. Cells expressing oncogenic PIK3CA mutants are indicated with red asterisks.

RNMT siRNA transfection (MCF7, HCC-1806, JIMT-1 and T47D) exhibited differences in RNMT expression and protein synthesis compared with those which are insensitive (IMEC, BT-549, MDA-MB-231, CAMA-1 and ZR-75-1) (figure 3). In all figures, the breast cancer cell lines are presented in order of sensitivity to RNMT reduction, MCF7 being the most sensitive and ZR-75-1 being the least sensitive. IMECs were analysed as a control cell line.

We investigated whether cellular dependency on RNMT correlated with RNMT expression or activity, measuring both basal levels and levels following RNMT siRNA transfection. RNMT and RAM expression were analysed by western blot performed on four independent samples (figure 3a,b). Expression of RNMT and RAM varied little across the cell panel, and a correlation between RNMT and RAM expression and dependency on RNMT was not observed. Cellular N7 cap guanosine methyltransferase activity, which is dependent on RNMT expression, also did not correlate with dependency on RNMT (figure 3c,d) [27]. For each cell line, the mRNA cap methyltransferase

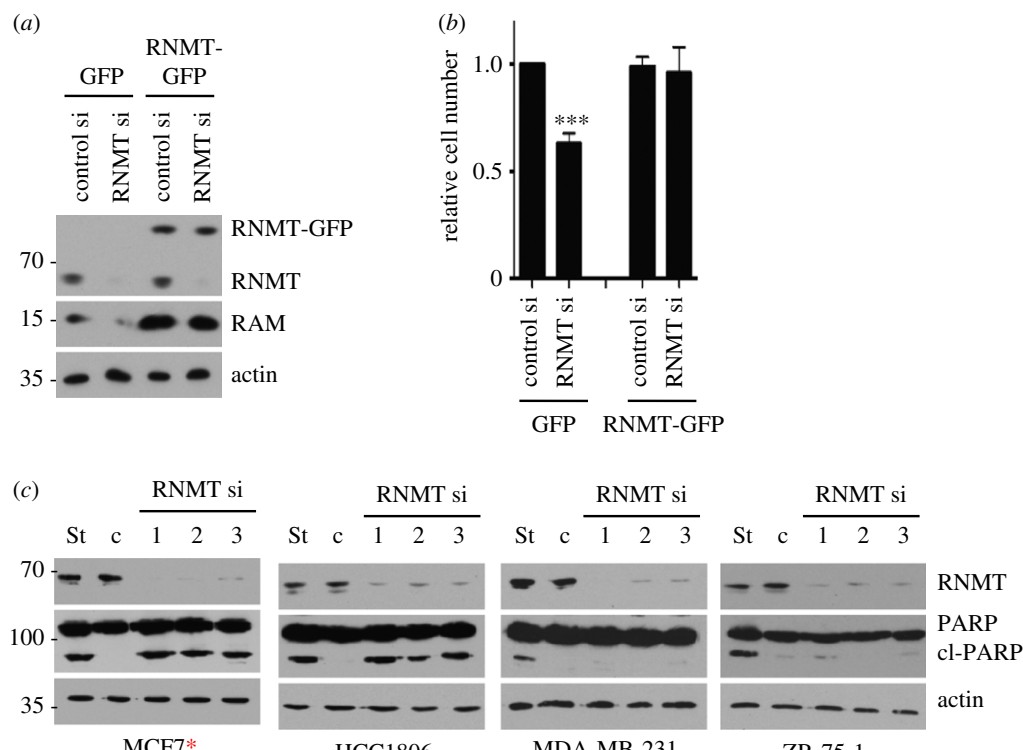

**Figure 2.** RNMT dependency correlates with increased apoptosis in breast cancer cell lines. (a) HCC1806 cells expressing siRNA-resistant RNMT-GFP or GFP control were transfected with RNMT siRNA1 or non-targeting siRNA. After 72 h, the resultant expression of RNMT-GFP, RNMT, RAM and actin was analysed by western blot. (b) Cell number relative to GFP-expressing control cells was determined, and statistical significance was calculated as above. (c) The cell lines indicated were transfected with three independent RNMT siRNAs. After 96 h, RNMT, PARP, cleaved PARP (cl PARP) and actin were detected by western blot. Cells were treated with staurosporine (St) to generate a positive control for PARP cleavage. Cells expressing oncogenic PIK3CA mutants are indicated with a red asterisk.

activity assay was performed on titration of cell extracts to determine the linear range of the assay (electronic supplementary material, figure S2). In this assay, a guanosine-capped transcript was incubated with cell extracts and the methyl donor, SAM (S-adenosyl methionine). Following incubation, the proportion of transcripts with an N7 methyl-guanosine cap (m7G(5′)ppp(5′)X) was determined by thin layer chromatography. Across the cell panel, only two cell lines exhibited significantly different N7 cap guanosine methyltransferase activity compared with IMEC; however, one was a cell line sensitive to RNMT siRNA transfection (HCC-1806), and the other was insensitive (CAMA-1) (figure 3c). Furthermore, the reduction in N7 cap guanosine methyltransferase activity following RNMT siRNA transfection did not vary significantly across the cell panel (figure 3d). All cell lines exhibited a 40–60% reduction in N7 cap guanosine methyltransferase activity in response to RNMT siRNA transfection, consistent with RNMT being equivalently depleted (figure 1c).

Given that RNMT–RAM expression is required for efficient cap-dependent translation, a reasonable hypothesis was that cells with higher protein synthesis or proliferation rates are more sensitive to loss of RNMT. An approximation of the protein synthesis rate was determined by measuring the rate of radio-labelled amino acid incorporation into cellular proteins. Although HCC1806, JIMT-1, CAMA-1 and ZR-75-1 cells exhibited higher basal net protein synthesis rates than IMEC, this did not correlate with dependency on RNMT (figure 3e). The reduction in protein synthesis in response to RNMT siRNA transfection also did not vary significantly across the cell panel (figure 3f). Furthermore, basal cell doubling time did not correlate with sensitivity to a reduction in RNMT (figure 3g).

## 3.3. RNMT dependency is induced by expression of oncogenic PIK3CA mutants

Three of the breast cancer cell lines sensitive to a reduction in RNMT have oncogenic hotspot activating PIK3CA mutations; MCF7 expresses PIK3CA E545 K, T47D expresses PIK3CA H1047R, and JIMT-1 expresses PIK3CA C420R (table 1; electronic supplementary material, figure S1). Conversely, all of the cell lines insensitive to RNMT inhibition expressed PIK3CA WT. No other cancer-associated mutations in the cell panel correlated with sensitivity to RNMT depletion. PIK3CA encodes the p110α subunit of the heterodimeric lipid kinase, PI3 K, which, upon stimulation by growth factors, activates downstream kinases including AKT and PDK1 (table 1). These effector kinases regulate major processes, including cell proliferation, growth and survival [28,29]. PIK3CA mutations are found in approximately 35% of breast cancer patients, with C420R, E545K and H1047R being the most common variants [23,30,31]. No other cancer-associated mutations in the cell panel correlated with sensitivity to RNMT depletion. Oncogenic PIK3CA mutants can confer growth advantage and render cells exquisitely sensitive to inhibition of PI3Kα [32–34]. Indeed, MCF7, T47D and JIMT-1 cell lines were found to have the highest sensitivity to the PI3Kα inhibitor, BYL719, in the breast cancer cell line panel, demonstrating that these cell lines require PI3Kα for proliferation and survival [35] (electronic supplementary material, figure S3).

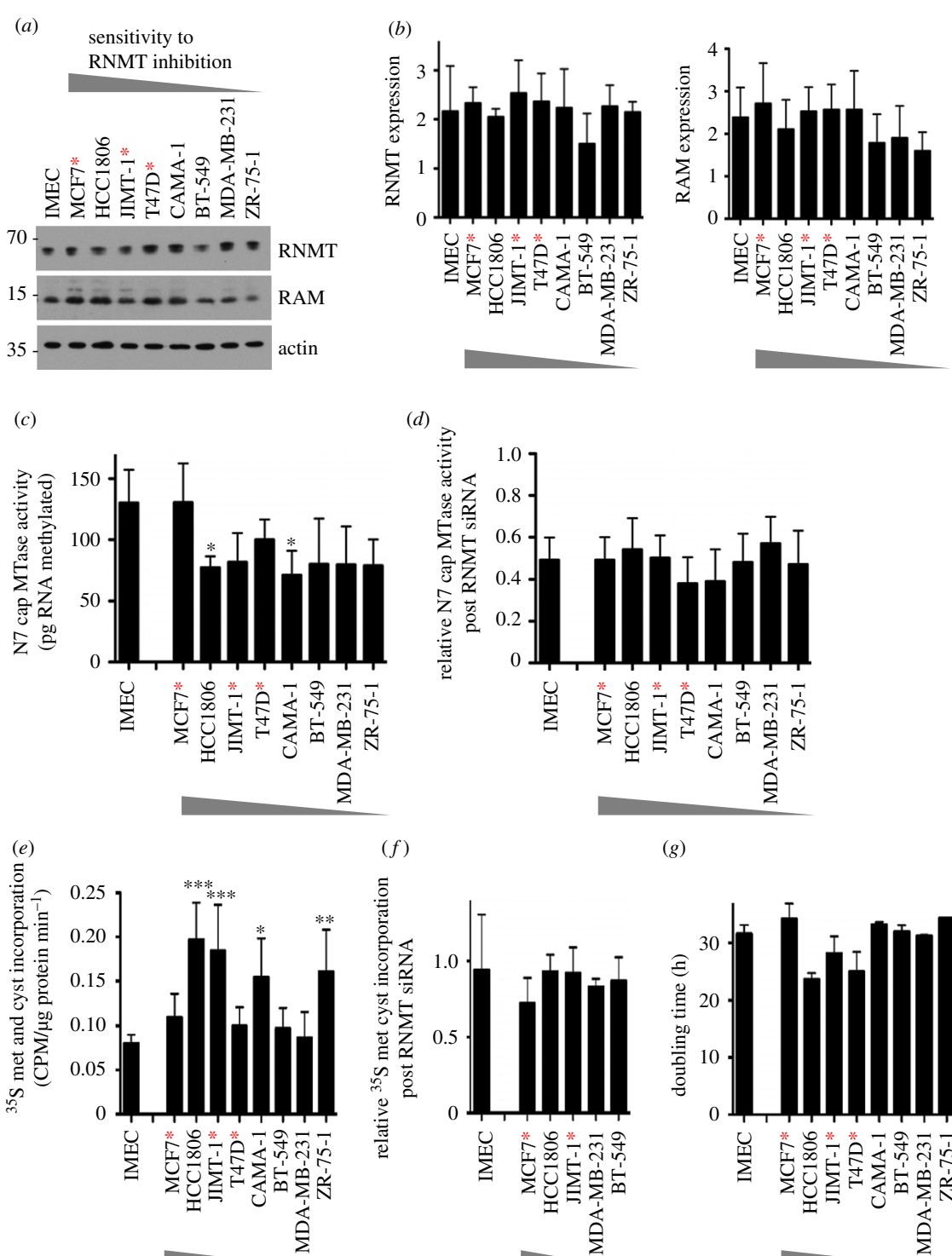

**Figure 3.** RNMT dependency does not correlate with RNMT expression or activity, the rate of protein synthesis or the rate of cell proliferation. (a) Levels of RNMT and RAM in log-phase cell extracts were analysed by western blot. Blots presented are representative of four independent experiments. (b) RNMT and RAM band intensities were quantified using IMAGEJ software. The bar charts depict average band intensity and standard deviation for four independent experiments. (c) N7 guanosine cap methyltransferase activity was determined in cell extracts using an *in vitro* assay. The chart depicts the average cap methyltransferase activity and standard deviation for four independent experiments. (d) N7 guanosine cap methyltransferase activity was determined in cell extracts 48 h after transfection with RNMT siRNA1 and expressed relative to values in control cells. (e) The rate of $^{35}$S methionine and $^{35}$S cysteine incorporation into cellular proteins was determined for log-phase cells. The chart depicts the mean and standard deviation for incorporation of amino acids per μg of cellular protein per min of assay reaction, for three independent experiments. (f) The rate of amino acid incorporation was determined in cells 48 h after transfection with RNMT siRNA1 and expressed relative to values in control cells. (g) The doubling time of log-phase cell lines determined. For charts, statistical significance was assessed using ANOVA followed by Dunnett's multiple comparison tests in comparison with IMEC values: *$p \leq 0.05$; **$p \leq 0.01$; ***$p \leq 0.001$. Cells expressing oncogenic PIK3CA mutants are indicated with red asterisks.

To determine whether oncogenic PIK3CA mutations sensitize cells to loss of RNMT, Myc-tagged PIK3CA WT, C420R, E545K, H1047R and vector control were introduced into ZR-75-1 cells by retroviral infection. ZR-75-1 cells expressed

WT PIK3CA and were demonstrated previously to be insensitive to RNMT siRNA transfection (figures 1 and 2). The expression of PI3 K p110α WT (product of PIK3CA gene) and mutants was confirmed in ZR-75-1 cells by the detection

royalsocietypublishing.org/journal/rsob Open Biol. 9: 190052

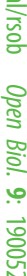

**Figure 4.** Expression of oncogenic PI3K p110α mutants C420, E545K and H1047R increases RNMT dependency in ZR-75-1 cells. (a) Myc-tagged PI3K p110α WT and mutants (C420R, E545 K and H1047R) were expressed in ZR-75-1 cells and detected by 9E10 antibody (anti-Myc tag) in western blots. (b) AKT Thr 308 and Ser 473 phosphorylation was analysed by western blot in the same cell extracts. (c) ZR-75-1/PI3K p110α cell lines were transfected with RNMT siRNA1 or non-targeting control for 48 h. Western blots were performed to detect RNMT. (d) Following transfection with siRNA, the cell number was determined over 3 days. Average cell number and standard deviation from three independent transfections are reported. (e) Seventy-two hours after transfection with RNMT siRNA 1 and 2, the cell number was determined relative to cells transfected with non-targeting siRNA. The average result and standard deviation for three independent experiments are depicted. The statistical significance was assessed by ANOVA followed by Dunnett's multiple comparison tests. '***' indicates a *p*-value of less than 0.001 compared with cells transfected with control siRNA.

of p110α (figure 4a). The activity of oncogenic p110α mutants was confirmed by the detection of increased AKT Thr 308 and Ser 473 phosphorylation, an established downstream output of PI3K signalling (figure 4b). As expected, ZR-75-1 cells expressing oncogenic PIK3CA mutants exhibited increased sensitivity to the p110α inhibitor, BYL719 [36] (electronic supplementary material, figure S4). Expression of RNMT was reduced in the ZR-75-1 cell lines by siRNA transfection (figure 4c). Basal RNMT expression and knockdown efficiency were equivalent in all cell lines. The proliferation rate of ZR-75-1 cells did not alter in response to exogenous

expression of PIK3CA WT or mutants (figure 4d). However, expression of the PIK3CA mutants increased the sensitivity of ZR-75-1 cells to reduced RNMT expression (figure 4d, lower chart). In electronic supplemental material, figure S5, data from both charts from figure 4d are presented in the same chart to illustrate that expression of PIK3CA mutants does not significantly alter the proliferation rate. Furthermore, RNMT siRNA transfection does not alter the proliferation rate in cells expressing PIK3CA WT, but does reduce the proliferation rate in cells expressing PIK3CA oncogenic mutants. Transfection of two independent RNMT

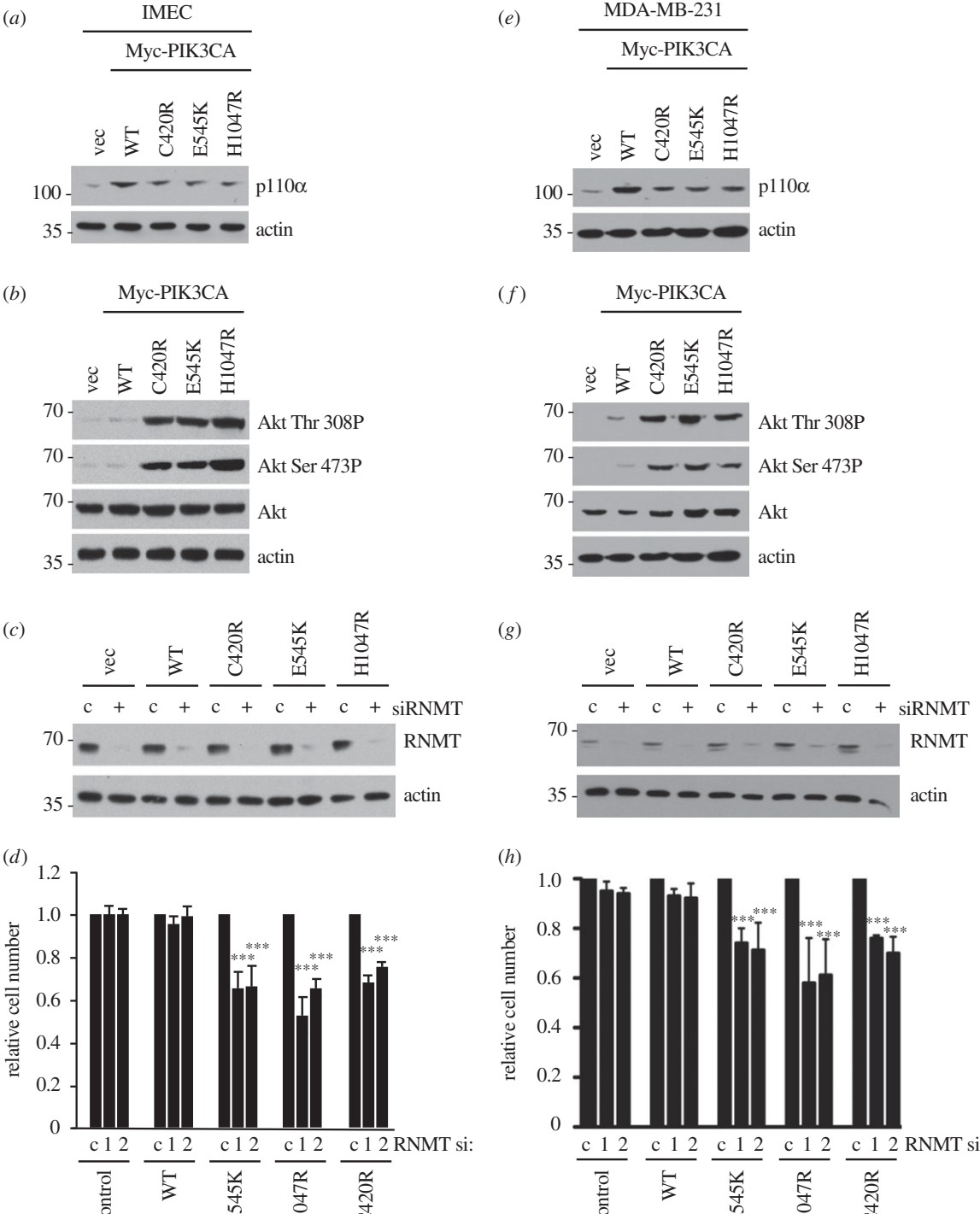

**Figure 5.** Expression of PI3K p110α C420R, E545K and H1047R increases RNMT dependency in IMEC and MDA-MB-231 cells. Data from IMEC presented in (*a*–*d*) and from MDA-MB-231 cells presented in (*e*–*h*). (*a,e*) Myc-tagged PI3K p110α WT and mutants were expressed in cells. PI3K p110α was detected in cell extracts by western blot. (*b,f*) AKT Ser 473 and Thr 308 phosphorylation was detected by western blot. (*c,g*) Cell lines were transfected with RNMT siRNA 1 or non-targeting control for 48 h. RNMT was detected by western blot. (*d,h*) Seventy-two hours after transfection with RNMT siRNA 1 and RNMT siRNA 2, the cell number was determined relative to cells transfected with control siRNA. Average and standard deviation for three independent experiments are depicted. Statistical significance was assessed by ANOVA followed by Dunnett's multiple comparison tests. '***' indicates a *p*-value of less than 0.001 compared with non-targeting control.

siRNAs caused a significant decrease in cell number in cells expressing oncogenic PIK3CA mutants (figure 4*e*).

To determine whether the expression of oncogenic PIK3CA mutants increases RNMT dependency in additional cell lines, PIK3CA WT, C420R, E545K, H1047R and vector control were expressed in the non-transformed mammary epithelial cell line IMEC and the breast cancer cell line MDA-MB-231 (figure 5*a,e*). Both cell lines expressed WT

PIK3CA and were demonstrated to be insensitive to RNMT siRNA transfection (figures 1 and 2). As expected, the oncogenic mutants increased PI3K signalling, as assessed by AKT phosphorylation (figure 5*b,f*). As with ZR-75-1 cells, expression of oncogenic PIK3CA mutants did not affect RNMT expression or the efficacy of RNMT siRNA (figure 5*c,g*). However, in both cell lines investigated, transfection of two independent RNMT siRNAs caused a

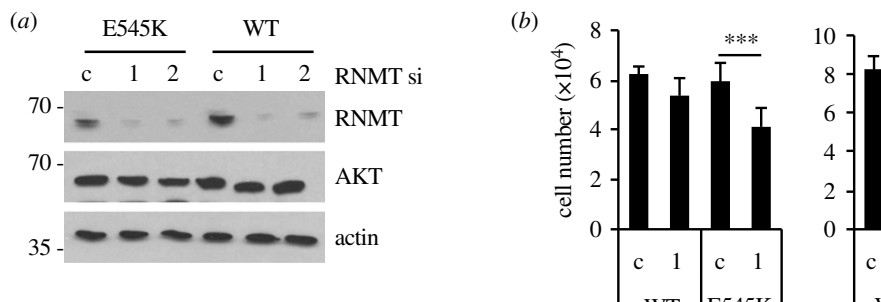

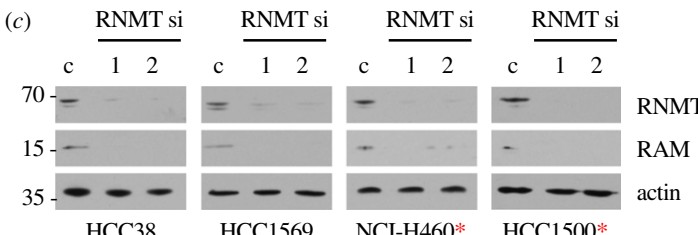

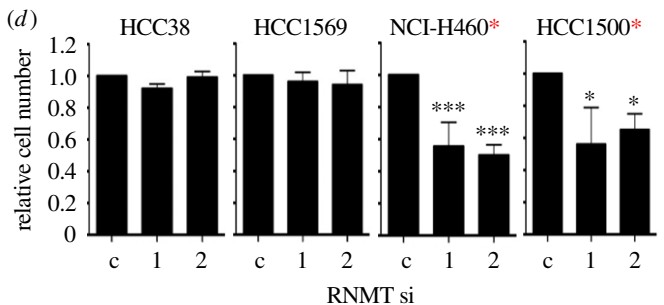

**Figure 6.** Expression of PI3K p110a E545K in isogenic human mammary epithelial cells increases dependency on RNMT. (*a*) Isogenic human mammary epithelial cells with WT or E545K activating mutation in the PIK3CA gene were transfected with two independent RNMT siRNAs or non-targeting control siRNA for 72 h. Western blot analysis was performed. (*b*) Cell number was determined relative to cells transfected with control siRNA. Statistical significance was assessed using ANOVA followed by Tukey's multiple comparison tests. *$p \leq 0.05$; **$p \leq 0.01$; ***$p \leq 0.001$. (*c*) Cancer cell lines HCC38, HCC1569, NCI-H460 and HCC1500 were transfected with two independent RNMT siRNAs or a non-targeting control siRNA for 96 h. Western blot analysis was performed. (*d*) Cell number was determined relative to cells transfected with control siRNA. Statistical significance was assessed using ANOVA followed by Dunnett's multiple comparison tests, as above. Cells expressing oncogenic PIK3CA mutants are indicated with a red asterisk.

significant decrease in cell number in cells expressing p110a C420R, E545K or H1047R (figure 5*d,h*; electronic supplementary material, figure S6a,b).

To extend these findings, similar experiments were performed using isogenic human mammary epithelial cells in which an endogenous copy of the PIK3CA gene was replaced by either PIK3CA E545K or WT control. The benefit of these cell lines is that the PIK3CA mutant is expressed at endogenous levels in an untransformed mammary epithelial cell line. RNMT expression was reduced by transfection of two independent siRNAs resulting in a similar level of knockdown (figure 6*a*). Following RNMT reduction, there was a significant reduction in p110α E545K-expressing cells but not in p110α WT-expressing cells (figure 6*b*).

To extend the finding that oncogenic PIK3CA mutants sensitize cells to a reduction in RNMT, RNMT knockdown was performed in three more breast cancer cell lines (HCC38, HCC1569 and HCC1500) and one lung carcinoma cell line (NCI-H460). The two cell lines with WT PIK3CA (HCC38 and HCC1569) were insensitive to a reduction in RNMT, whereas the two cell lines with mutations in PIK3CA (NCI-H460 and HCC1500) exhibited reduced proliferation in response to RNMT siRNA transfection (figure 6*c,d*). This is

consistent with cancer cells harbouring oncogenic PIK3CA mutations exhibiting enhanced dependency on RNMT.

## 3.4. Inhibition of PI3K signalling reduces RNMT dependency

To further investigate the hypothesis that oncogenic PI3K p110α signalling increases dependency on RNMT, we used inhibitors of PI3Kα activity. Phospho-AKT 473 served as a readout of PI3K inhibitor target engagement (figure 7*a,c*). GDC-0941 is a pan PI3K inhibitor which inhibits p110 α, β, δ and γ [37] (figure 7*a*). As expected, reducing RNMT expression in vehicle-treated T47D and MCF-7 cells significantly reduced cell proliferation (figure 7*b*). However, incubation with a sub-lethal dose of GDC-0941 suppressed this proliferative defect. BYL719 selectively inhibits p110α [37] (figure 7*c*). As expected, in vehicle-treated T47D and JIMT cells reducing RNMT expression resulted in reduced proliferation (figure 7*d*). However, incubation with a sub-lethal dose of BYL719 suppressed this proliferative defect. Therefore, suppression of PI3K p110a activity desensitizes these cell lines to RNMT inhibition. Collectively, these results

royalsocietypublishing.org/journal/rsob    Open Biol. 9: 190052

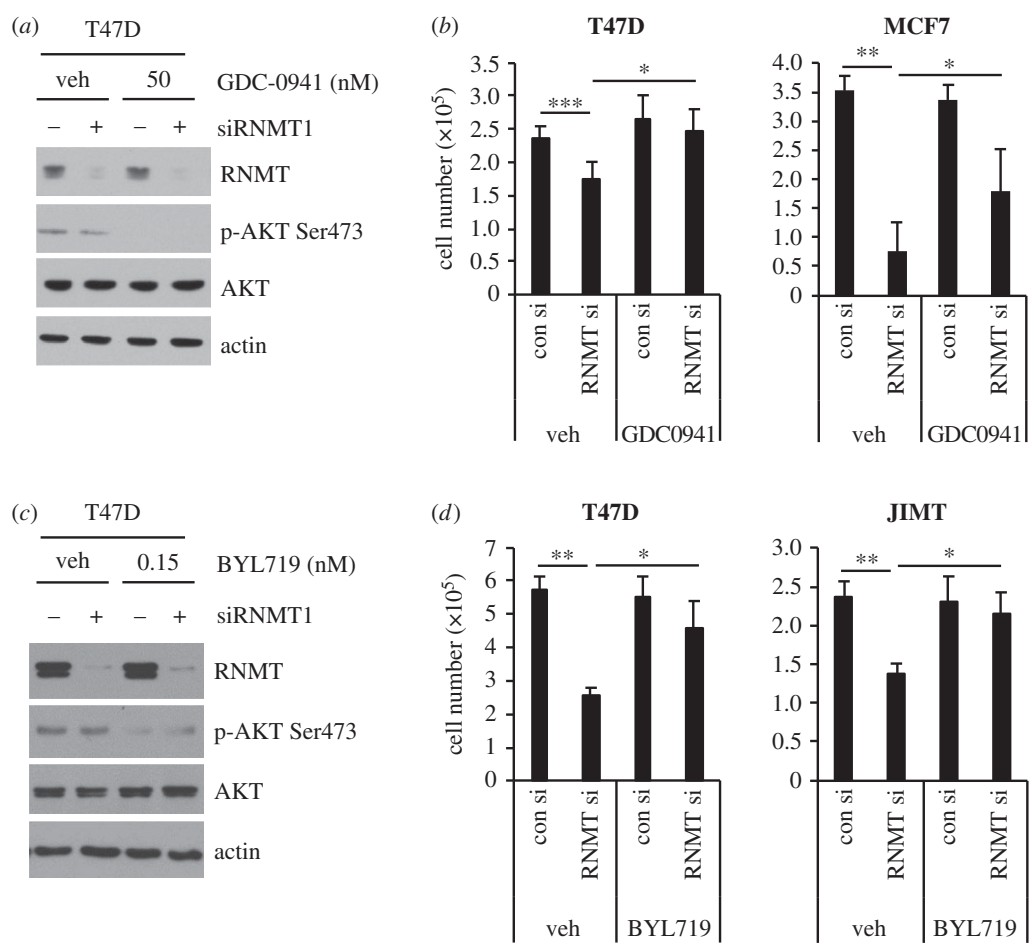

**Figure 7.** Pharmacological inhibition of PI3Kα reduces RNMT dependency. (a,b) T47D and MCF7 cells were incubated with 50 nM GDC-0941 or vehicle control (veh), and transfected with RNMT siRNA1 for 72 h. (c,d) T47D and JIMT cells were incubated with 15 nM BYL719 or vehicle control (veh), and transfected with RNMT siRNA1 for 72 h. (a,c) Cell extracts were analysed by western blot. (b,d) Average cell number and standard deviation for at least three independent experiments are presented. Statistical significance was assessed using ANOVA followed by Dunnett's multiple comparison tests. $*p \leq 0.05$; $**p \leq 0.01$; $***p \leq 0.001$ indicates $p$-values relative to control indicated.

demonstrate that PI3K p110α signalling contributes to RNMT dependency in breast cancer cell lines expressing oncogenic PIK3CA mutants.

## 4. Discussion

Breast cancer is the most prevalent cancer in women worldwide, causing over 500 000 deaths annually [35,38]. Although the development of hormone and HER2-targeted therapies has greatly improved the prognosis of patients with luminal A/B and HER2 breast cancer, respectively, the emergence of resistance remains a major problem, limiting the effectiveness of drug treatment. Moreover, currently, there are limited treatment options for triple negative breast cancer, which represent 10–20% of breast cancer cases [39–41]. The PI3K–AKT–mTOR pathway is a growth and survival pathway that is frequently deregulated in breast cancer due to genetic alterations in the pathway, including activating mutations in the oncogene PIK3CA and deletion of the tumour suppressor PTEN. Consequently, several inhibitors of this pathway are currently in clinical development for the treatment of breast cancer [42,43].

Among new targets being explored for the treatment of cancer are those that target the basic mechanisms of gene expression [1,2]. Deregulation of common oncogenes (e.g.

c-Myc, PIK3CA, Ras and Raf) and loss of common tumour suppressors (e.g. p53, PTEN and SWI/SNF) are associated with deregulated or enhanced gene expression. Since the formation of the mRNA cap is a key step in gene expression, we considered the mRNA cap methyltransferase, RNMT, which completes the basic functional mRNA cap, as a therapeutic target. Specifically, we were interested in identifying oncogenes or tumour suppressors, which sensitize cells to RNMT inhibition. The aim was to determine which breast cancer subtypes or gene signatures may be responsive to therapies targeting RNMT.

To address whether different breast cancer cell lines exhibit differential sensitivity to inhibition of RNMT, we reduced RNMT expression in a panel of breast cancer cell lines and non-transformed cells by transfection of RNMT siRNA (specific inhibitors of RNMT are not currently available). Using this technique, we reduced cellular cap methyltransferase activity by 50–60% across the cell panel (figure 3d). We observed that a subset of breast cancer cell lines exhibits enhanced sensitivity to inhibition of RNMT, compared with a non-transformed control cell line (figures 1 and 6). This differential dependency on RNMT did not correlate with RNMT expression level or activity, both in log-phase cells and following RNMT siRNA transfection (figures 1 and 3). However, oncogenic mutations in the gene PIK3CA, which encodes the p110α catalytic subunit of PI3K, were found in

five out of six cancer cell lines exhibiting enhanced sensitivity to RNMT inhibition (figures 1 and 6, and table 1). Of note, breast cancer cell lines with PTEN loss (CAMA-1, BT-549 and ZR-75-1) were not sensitive to RNMT inhibition, suggesting that PIK3CA mutations, rather than PTEN loss, promotes RNMT dependency in breast cancer. This observation is consistent with an emerging body of data, suggesting that genetic alterations in different parts of the PI3K pathway have distinct biological effects [44–46]. HCC1806 cells do not have a PIK3CA mutation but do have an LKB1 deletion. As LKB1 indirectly suppresses mTOR signalling via AMPK, LKB1 deletion increases mTOR signalling, as does mutant PIK3CA via AKT signalling. Although the mTOR signalling pathway is unlikely to provide a complete explanation as to why HCC1806 cells are sensitive to RNMT inhibition, the LKB1 deletion may contribute to the phenotype.

Strengthening the link between PIK3CA mutations and RNMT inhibition, breast cancer cell lines and mammary epithelial cells, which were insensitive to RNMT siRNA transfection, could be rendered sensitive by the expression of PIK3CA oncogenic mutants (figures 4 and 5). Conversely, breast cell lines with endogenous oncogenic PIK3CA mutations were desensitized to RNMT repression by the use of PI3Kα inhibitors (figure 7), suggesting that oncogenic PIK3CA mutations promote RNMT dependence through increased PI3Kα signalling. Collectively, these data indicate that breast cells harbouring oncogenic PIK3CA mutations have enhanced dependence on RNMT (mRNA cap methylation) for proliferation and survival, making RNMT a promising therapeutic target in PIK3CA mutant breast cancer. Since approximately 35% of all breast cancers contain mutations in PIK3CA, these findings could have a significant impact.

## 4.1. Why do oncogenic PIK3CA mutations increase dependency on RNMT?

The mRNA cap protects transcripts from degradation and recruits protein complexes involved in RNA processing and translation [47,48]. RNMT also has a major role in enhancing the transcription of the most highly expressed genes [11]. In previous studies in which levels of RNMT or its activator RAM were altered, genes responded differentially at the mRNA level and translation rate [11,14,19,49,50]. This results in cellular proteins and signalling pathways being differentially dependent on RNMT for expression. Why are genes differentially dependent on RNMT? Some transcripts may be dependent on high concentrations of RNMT to be methylated correctly. Certain guanosine-capped transcripts may have limited access to RNMT, because of the chromatin context, the speed of transcription or RNA secondary structure. Alternatively, some transcripts may not require their mRNA cap to be methylated on the cap guanosine, such as by using IRES (internal ribosome entry site)-dependent translation initiation. Although RNMT-catalysed methylation of the mRNA cap guanosine increases affinity for cap-binding complexes, it is not absolutely required. Some transcripts may be largely independent of the cap guanosine methylation for translation initiation. In experiments performed here, when cellular RNMT activity was reduced by approximately 50%, the global translation rate (approximated by amino acid incorporation into proteins) was reduced by

only 5–10% in most cell lines investigated, indicating that RNMT is present in excess for the expression of most genes (figure 3f).

The mechanism by which oncogenic PIK3CA increases dependency on RNMT remains an open question. Oncogenic PIK3CA mutants (and downstream activation of mTORC1-driven cap-dependent translation) may alter the expression of proteins which render cells sensitive to RNMT inhibition. Alternatively, RNMT inhibition may affect protein expression similarly in all breast cancer cell lines, regardless of PIK3CA status; cells expressing PIK3CA may have increased dependency on a subset of RNMT-dependent proteins as a mechanism of oncogene addiction. We investigated whether the expression or phosphorylation of direct or indirect AKT substrates, which are involved in mRNA translation, were differentially expressed in IMECs and the breast cancer cell lines (electronic supplementary material, figure S7). The expression and phosphorylation level of AKT, eIF4E BP-1 and P70-S6 kinase did not correlate with sensitivity to RNMT suppression. We also investigated whether expression of the p85 and p110 subunits of PI3K were supressed in response to RNMT siRNA transfection, in MCF7 and HCC1806 (sensitive to RNMT siRNA transfection), and MDA-MB-231 and ZR-75-1 (insensitive to RNMT siRNA transfection). Expression of p85 and p110 was not robustly up or downregulated following 48 and 72 h RNMT siRNA transfection (electronic supplementary material, figure S8a). This experiment was also repeated with three independent RNMT siRNAs with similar results (not shown). This suggests that PIK3CA oncogenic mutations increase dependency on RNMT independently of increasing translation of p85 and p110.

In HeLa cells and other lines, repression of capping enzymes results in repression of c-Myc expression and RNA pol II phosphorylation [11,15]. c-Myc is induced by mitogenic signalling (including via the PI3K pathway) to promote cell proliferation, including increasing RNA pol II phosphorylation, which reflects global transcription levels. We therefore investigated whether, in the cell lines sensitive to RNMT siRNA transfection, c-Myc and/or RNA pol II phosphorylation is repressed when RNMT expression is reduced. In MCF7, HCC-1806, JIMT-1 and T47D cells, c-Myc expression and RNA pol II phosphorylation were not consistently repressed in response to RNMT siRNA transfection (electronic supplementary material, figure S8b). Deep proteomic analyses of a large panel of breast carcinoma cell lines will be useful in identifying the full spectrum of RNMT-dependent proteins in PIK3CA-mutant breast cancer.

In this study we provide evidence to suggest RNMT should be explored as a therapeutic target in breast cancers harbouring PI3K p110α mutations. The development of specific tool compounds, which inhibit RNMT function and the elucidation of the function of RNMT in vivo, will be important next steps in these investigations.

Data accessibility. This article has no additional data.

Authors' contributions. S.D., O.L., R.L. and V.H.C. made substantial contributions to the conception of the project, acquisition of data, analysis of data and interpretation of data. All authors were involved in drafting and revising the article, and approving the article.

Competing interests. The authors have no competing interests to declare.

Funding. This research was funded by a Medical Research Council Senior Fellowship (MR/K024213/1), a Royal Society Wolfson

Research Merit Award (WRM\R1\180008), a Lister Institute Prize Fellowship, a BBSRC PhD studentship, a Wellcome Trust Centre Award (097945/Z/11/Z) and a Wellcome Trust Strategic Award (100476/Z/12/Z)

Acknowledgements. We thank members of the Dundee Drug Discovery Unit and Cowling and Alessi labs for discussions, and Simone Weidlich, Division of Signal Transduction Therapies, the University of Dundee for cloning assistance.

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
