## [Reviewer comments · Open Biology]

Review History

RSOB-18-0144.R0 (Original submission)

Review form: Reviewer 1

Recommendation

Major revision is needed (please make suggestions in comments)

Are each of the following suitable for general readers?

- a) **Title**
Yes
- b) **Summary**
Yes
- c) **Introduction**
Yes

Is the length of the paper justified?

Yes

Should the paper be seen by a specialist statistical reviewer?

No

Is it clear how to make all supporting data available?

Yes

Is the supplementary material necessary; and if so is it adequate and clear?

Yes

Do you have any ethical concerns with this paper?

No

Comments to the Author

This is a well structured study with clear description and illustration of results. The data reported are novel and potentially interesting.

However, there are two major issues to be addressed:

- The selection of cell line models was a bit random as also stated by the authors. Perhaps a more appropriate selection could have stemmed from an analysis to select based on RNMT expression.
- Most importantly, the authors based their assessment for mutations based on the COSMIC database, assuming that the reported mutation were present in the cells used for experiments. This is suboptimal, as all of the data stand on this in silico analysis. The authors should perform mutational analysis on the cells used, this is needed, at least for all exons of the PIK3CA gene. It is also unclear how the paper is only focused on PIK3CA mutations. The authors extracted data from COSMIC, it is not clear why then results are only reported for PIK3CA, as the authors picked this mutation over others. What's the rationale behind that? This needs to be contextualized. It is unlikely this gene was the only correlated to RNMT expression. Without these data and further explanation the paper is not suitable for publication

Review form: Reviewer 2

Recommendation

Accept with minor revision (please list in comments)

Are each of the following suitable for general readers?

- a) **Title**
Yes
- b) **Summary**
Yes
- c) **Introduction**
Yes

Is the length of the paper justified?

Yes

Should the paper be seen by a specialist statistical reviewer?

No

Is it clear how to make all supporting data available?

Yes

Is the supplementary material necessary; and if so is it adequate and clear?

Yes

Do you have any ethical concerns with this paper?

No

Comments to the Author

The manuscript is very well written, and generally quite clear. Some clarifications would be helpful:

1) In Figure 1, the authors show that RAM protein levels decrease when RNMT is knocked-down with RNA interference. They reference an article in which this has previously been reported but do not comment further. Is it known how this is mediated? A comment on this phenomenon would be useful for the reader in the text.

Importantly, in Fig 1A-C it would be useful to carry out these experiments with an RNAi resistant RAM expressing cell line to separate the effect on cell growth etc from the off-target loss of RAM (similar to the experiment in Fig2A)

2) Can the authors comment on the growth reduction in HCC1806 cells with wild-type PIK3CA? Why does it behave more like the mutant PIK3CA cell lines?

3) In Figure 4D, these data should be on one plot. Are both si-Control data the same? Does overexpression of PIK3CA mutants give a growth advantage beyond the wild-type?

4) In Fig 5D&H as well as 6B, it would be useful to know the raw non-normalized cell numbers to compare mutant overexpression to wild-type

5) In the discussion the authors state that 'We investigated PI3K signalling proteins extensively, but did not find that transfection of RNMT siRNA altered their expression or phosphorylation (not shown)' - It would be useful to show this data. Which proteins were investigated?

Decision letter (RSOB-18-0144.R0)

05-Oct-2018

Dear Professor Cowling:

We are writing to inform you that the Editor has reached a decision on your manuscript RSOB-18-0144 entitled "Oncogenic PIK3CA mutations increase dependency on the mRNA cap methyltransferase, RNMT, in breast cancer cells", submitted to Open Biology.

As you will see from the reviewers' comments below, there are a number of criticisms that prevent us from accepting your manuscript at this stage. The reviewers suggest, however, that a

revised version could be acceptable, if you are able to address their concerns. If you think that you can deal satisfactorily with the reviewer's suggestions, we would be pleased to consider a revised manuscript.

The revision will be re-reviewed, where possible, by the original referees. As such, please submit the revised version of your manuscript within six weeks. If you do not think you will be able to meet this date please let us know immediately.

When submitting your revised manuscript, please respond to the comments made by the referee(s) and upload a file "Response to Referees" in "Section 6 - File Upload". You can use this to document any changes you make to the original manuscript. In order to expedite the processing of the revised manuscript, please be as specific as possible in your response to the referee(s).

Please see our detailed instructions for revision requirements
<https://royalsociety.org/journals/authors/author-guidelines/>

Sincerely,

The Open Biology Team
mailto: openbiology@royalsociety.org

ditage Insights by clicking on the following link: <https://www.surveymonkey.com/r/author-perspectives-on-academic-publishing-royal-society>
This should take no more than 15 minutes and you will have the opportunity to enter a prize draw. We hope these results will provide us with valuable insights we can use to improve our service.

Editor's Comments to Author(s):
Please address all comments of the referees.

Reviewer(s)' Comments to Author(s):

Referee: 1

Comments to the Author(s)

This is a well structured study with clear description and illustration of results. The data reported are novel and potentially interesting.

However, there are two major issues to be addressed:

- The selection of cell line models was a bit random as also stated by the authors. Perhaps a more appropriate selection could have stemmed from an analysis to select based on RNMT expression.

- Most importantly, the authors based their assessment for mutations based on the COSMIC database, assuming that the reported mutation were present in the cells used for experiments. This is suboptimal, as all of the data stand on this in silico analysis. The authors should perform mutational analysis on the cells used, this is needed, at least for all exons of the PIK3CA gene. It is also unclear how the paper is only focused on PIK3CA mutations. The authors extracted data from COSMIC, it is not clear why then results are only reported for PIK3CA, as the authors picked this mutation over others. What's the rationale behind that? This needs to be contextualized. It is unlikely this gene was the only correlated to RNMT expression. Without these data and further explanation the paper is not suitable for publication

Referee: 2

Comments to the Author(s)

The manuscript is very well written, and generally quite clear. Some clarifications would be helpful:

1) In Figure 1, the authors show that RAM protein levels decrease when RNMT is knocked-down with RNA interference. They reference an article in which this has previously been reported but do not comment further. Is it known how this is mediated? A comment on this phenomenon would be useful for the reader in the text.

Importantly, in Fig 1A-C it would be useful to carry out these experiments with an RNAi resistant RAM expressing cell line to separate the effect on cell growth etc from the off-target loss of RAM (similar to the experiment in Fig2A)

2) Can the authors comment on the growth reduction in HCC1806 cells with wild-type PIK3CA? Why does it behave more like the mutant PIK3CA cell lines?

3) In Figure 4D, these data should be on one plot. Are both si-Control data the same? Does overexpression of PIK3CA mutants give a growth advantage beyond the wild-type?

4) In Fig 5D&H as well as 6B, it would be useful to know the raw non-normalized cell numbers to compare mutant overexpression to wild-type

5) In the discussion the authors state that 'We investigated PI3K signalling proteins extensively, but did not find that transfection of RNMT siRNA altered their expression or phosphorylation (not shown)' - It would be useful to show this data. Which proteins were investigated?

Author's Response to Decision Letter for (RSOB-18-0144.R0)

See Appendix A.

RSOB-19-0052.R0

Review form: Reviewer 1

Recommendation

Accept as is

Are each of the following suitable for general readers?

- a) **Title**
Yes
- b) **Summary**
Yes
- c) **Introduction**
Yes

Is the length of the paper justified?

Yes

Should the paper be seen by a specialist statistical reviewer?

No

Is it clear how to make all supporting data available?

Not Applicable

Is the supplementary material necessary; and if so is it adequate and clear?

Yes

Do you have any ethical concerns with this paper?

No

Comments to the Author

N/A

Decision letter (RSOB-19-0052.R0)

19-Mar-2019

Dear Professor Cowling

We are pleased to inform you that your manuscript entitled "Oncogenic PIK3CA mutations increase dependency on the mRNA cap methyltransferase, RNMT, in breast cancer cells" has been accepted by the Editor for publication in Open Biology.

You can expect to receive a proof of your article from our Production office in due course, please

check your spam filter if you do not receive it within the next 10 working days. Please let us know if you are likely to be away from e-mail contact during this time.

Article processing charge

Please note that the article processing charge is immediately payable. A separate email will be sent out shortly to confirm the charge due. The preferred payment method is by credit card; however, other payment options are available.

Sincerely,

The Open Biology Team
mailto:openbiology@royalsociety.org

Appendix A

Manuscript RSOB-18-0144 Response to reviewers

Referee: 1

Comments to the Author(s)

This is a well structured study with clear description and illustration of results. The data reported are novel and potentially interesting.

However, there are two major issues to be addressed:

-The selection of cell line models was a bit random as also stated by the authors. Perhaps a more appropriate selection could have stemmed from an analysis to select based on RNMT expression.

RESPONSE:

Thank you for reviewing our manuscript.

The breast cancer cell lines were chosen as having a spectrum of oncogenic mutations. We have now added this statement to line 172, "Initially a panel of eight breast cancer cell lines with a spectrum of mutations was analysed". The expression of RNMT and RAM was equivalent in all eight cell lines (Figure 3A and B). There are many strategies that one can take to select a panel of cell lines for the initial experiments performed here. Our strategy of choosing a spectrum of cell lines was sufficient to highlight the correlation between PIK3CA mutations and RNMT sensitivity. The role of PIK3CA in sensitising cells to RNMT suppression was validated by expression of PIK3CA oncogenic mutations (Figures 4-6) and use of PIK3CA inhibitors (Figure 7).

-Most importantly, the authors based their assessment for mutations based on the COSMIC database, assuming that the reported mutation were present in the cells used for experiments. This is suboptimal, as all of the data stand on this in silico analysis. The authors should perform mutational analysis on the cells used, this is needed, at least for all exons of the PIK3CA gene.

RESPONSE: In the revised manuscript we have sequenced the coding regions of PIK3CA gene. Line 186 "The PI3KCA coding region sequence was verified for all cell lines (Supplemental Figure 1)". Line 101, method of sequencing.

It is also unclear how the paper is only focused on PIK3CA mutations. The authors extracted data from COSMIC, it is not clear why then results are only reported for PIK3CA, as the authors picked this mutation over others. What's the rational behind that? This needs to be contextualized. It is unlikely this gene was the only correlated to RNMT expression.

RESPONSE: In Table 1, Line 176: "Known mutations of cancer-associated genes in these cell lines were extracted from the COSMIC database (Table 1)". In this table we presented all genes in COSMIC which are defined as cancer-associated. For example, for MCF7 cells we reported the following genes with oncogenic mutations: ATP2B3, EP300, ERBB4, FUS, GATA3, KMT2C, MAP3K13, MYH9, PIK3CA.

None of the oncogenic mutations correlated with RNMT expression, which is equivalent across the panel (Figure 3A and B). What was observed, as described in the results section, was that 3 out of 4 cell lines which are sensitive to RNMT siRNA transfection also carried PIK3CA mutations. 4 out of 4 cell lines insensitive to RNMT siRNA transfection had PIK3CA WT. No other oncogenic mutations correlated with RNMT-sensitivity equivalently to PIK3CA. Therefore we proceeded to investigate the role of PIK3CA mutations in conferring RNMT-sensitivity. Line 269, "Three of the breast cancer cell lines sensitive to a reduction in RNMT have oncogenic hotspot PIK3CA mutations; MCF7 expresses PIK3CA E545K, T47D expresses PIK3CA H1047R, and JIMT-1 expresses PIK3CA C420R (Table 1). Conversely, all of the cell lines insensitive to RNMT inhibition expressed PIK3CA WT."

Referee: 2

Comments to the Author(s)

The manuscript is very well written, and generally quite clear. Some clarifications would be helpful:

1) In Figure 1, the authors show that RAM protein levels decrease when RNMT is knocked-down with RNA interference. They reference an article in which this has previously been reported but do not comment further. Is it known how this is mediated? A comment on this phenomenon would be useful for the reader in the text.

RESPONSE:

Thank you for reviewing our manuscript.

In Line 197 we now clarify: "RNMT and RAM are co-translated and their interaction protects the two proteins from proteasome-mediated degradation". We also add 2 references to support this statement. {Gonatopoulos-Pournatzis, 2014 #578;Gonatopoulos-Pournatzis, 2011 #508}..

Importantly, in Fig 1A-C it would be useful to carry out these experiment with an RNAi resistant RAM expressing cell line to separate the effect on cell growth etc from the off-target loss of RAM (similar to the experiment in Fig2A).

RESPONSE: In cancer cell lines, repression of RNMT results in an equivalent repression of RAM (as seen in figure 1). Similarly, repression of RAM results in an equivalent loss of RNMT. These are not off-target effects but rather reflect the loss of the RNMT-RAM complex. Since RNMT and RAM stabilise each other, a reduction in RNMT leads to a loss of stabilisation of RAM, and visa versa.

RAM has never been isolated as a monomer and has only been found in a complex with RNMT – therefore RAM does not operate independently of RNMT. Therefore we believe that expressing RAM independently of RNMT does not have much value in understanding the biology in question. In addition we have found it technically impossible to express RNMT or RAM independently of the other for longer than a few hours (probably due to co-translational RNMT-RAM stability).

2) Can the authors comment on the growth reduction in HCC1806 cells with wild-type PIK3CA? Why does it behave more like the mutant PIK3CA cell lines?

RESPONSE: HCC1806 cells do not have a PIK3CA mutation, but do have an LKB1 deletion. LKB1 indirectly suppresses mTOR signalling. LKB1 deletion therefore increases mTOR signalling, as does mutant PIK3CA via AKT signalling. Although we do not think that this is the complete answer as to why HCC1806 cells are sensitive to RNMT inhibition, we think that LKB1 deletion probably contributes to the phenotype. We have added this speculation to the discussion (line 394).

3) In Figure 4D, these data should be on one plot. Are both si-Control data the same? Does overexpression of PIK3CA mutants give a growth advantage beyond the wild-type?

RESPONSE: We have now presented the data from figure 4D in one plot in supplemental figure 5, line 301. The reason we kept batches of curves separate in figure 4D is that it is very difficult to see the error bars if data is presented all together. As can be seen in supplemental figure 5, expression of PIK3CA mutants does not increase proliferation of ZR-75-1 under the conditions used here. RNMT siRNA transfection only reduces the proliferation rate when cells expression PIK3CA oncogenic mutants.

4) In Fig 5D&H as well as 6B, it would be useful to know the raw non-normalized cell numbers to compare mutant overexpression to wild-type

RESPONSE: This presentation of the data from figure 5d and h is now in supplemental figure 6 a and b. As can be seen in b, expression of the PIK3CA mutants slightly increases proliferation in MDA-MB-231 cell lines but not significantly over the time course of the experiment. In figure 6b, we now present the raw data rather than normalised figures.

5) In the discussion the authors state that 'We investigated PI3K signalling proteins extensively, but did not find that transfection of RNMT siRNA altered their expression or phosphorylation (not shown)' – It would be useful to show this data. Which proteins were investigated?

RESPONSE: We investigated whether the expression or phosphorylation of direct or indirect AKT substrates which are involved in mRNA translation, were differentially expressed in IMECs and the breast cancer cell lines (Supplemental Figure 7 and line 434). The expression and phosphorylation level of AKT, eIF4E BP-1 and P70-S6 kinase did not correlate with sensitivity to RNMT suppression.

We also investigated whether the expression of the p85 and p110 subunits of PI3K were suppressed in response to RNMT siRNA transfection, in MCF7 and HCC1806 (sensitive to RNMT siRNA transfection), and MDA-MB-231 and ZR-75-1 (insensitive to RNMT siRNA transfection). Expression of p85 and p110 was not robustly up or downregulated following 48 and 72 hours RNMT siRNA transfection (Supplemental Figure 8a, line 448). This experiment was also repeated with three independent RNMT siRNAs with similar results (not shown). In HeLa cells and other lines, repression of capping enzymes results in repression of c-Myc and RNA pol II phosphorylation {Lombardi, 2016 #931}{Varshney, 2018 #1250}. Since c-Myc promotes cell proliferation and RNA pol II phosphorylation reflects global transcription, we investigate whether in the cell lines sensitive to RNMT siRNA transfection, c-Myc and or RNA pol II phosphorylation is repressed in response to RNMT siRNA transfection. In MCF7, HCC-1806, JIMT-1 and T47D cells, in response to RNMT siRNA transfection, c-Myc expression and RNA pol II phosphorylation were not consistently repressed (Supplemental Figure 8b, line 462).